# Electroacupuncture alleviates pain-like behaviors through modulating DNMT3a/MOR signaling pathway in CCI rats

Feng Wang, Chengcheng Zhou, Chuangbo Xie, Haoyuan Wang, Xiangyu Li, Gaofeng Zhao ⓘD *

Department of Anesthesiology, Second Affiliated Hospital of Guangzhou University of Chinese Medicine, Guangzhou, Guangdong, China

* zhaogaofengzyy@163.com

## Abstract

Neuropathic pain (NP) affects mental health and social functioning of people. Electroacupuncture (EA) has been shown to be effective in relieving NP in clinical practice, but the specific mechanism is still unclear. In our study, we aimed to explore the mechanism of how EA relieving NP by chronic constriction injury (CCI) rat model. EA treatment was performed at acupoints *Zusanli* (ST36) and *Yanglingquan* (GB34) after CCI 1 week and after AAV dorsal root ganglion (DRG) injection 3 weeks. EA was performed 30 minutes per day for 7 days. The paw withdrawal threshold (PWT) and paw withdrawal latency (PWL) were checked. The expressions of DNA Methyltransferase 3 Alpha (DNMT3a), Mu opioid receptors (MOR), pain-related signals (Fos, ERK1/2, and pERK1/2) and glial cell activity-related factors (CD11b, Iba1, and GFAP) in spinal cord dorsal horn (SCDH) and DRG were detected by western blotting, quantitative polymerase chain reaction reverse transcription (RT-qPCR). The expressions of DNMT3a and MOR in SCDH and DRG were also observed by immunofluorescence experiments. Our results revealed that CCI increased DNMT3a and decreased MOR expressions in DRG, which could be reversed by EA. EA at Unilateral acupoints *Zusanli* (ST36) and *Yanglingquan* (GB34) increased the PWT as well as PWL of the hind paw of CCI rats by down-regulating the expression of pain-related signals (Fos, ERK1/2, and pERK1/2) and glial cell activity-related factors (CD11b, Iba1, and GFAP) in SCDH as well as DRG. EA therapy could produce favorable analgesic results in individuals who experienced pain sensitivity arising from peripheral nerve injury. EA may improve analgesia by increasing MOR expression through the inhibition of DNMT3a in DRG.

**Data availability statement:** All relevant data are within the manuscript and its Supporting information files.

**Funding:** This work was supported by the Natural Science Foundation of Guangdong Province [grant number 2020A1515010956].

**Competing interests:** The authors have declared that no competing interests exist.

**Abbreviations:** AAV, Adeno-associated virus; CCI, Chronic constriction injury; DNMT3a, DNA methyltransferase 3 Alpha; DNMTs, DNA methyltransferases; DRG, Dorsal root ganglion; EA, Electroacupuncture; GB34, Yanglingquan; MDB1, Methyl-CpG binding domain protein 1; MOR, Mu opioid receptor; NP, Neuropathic pain; PWL, Paw withdrawal latency; PWT, Paw withdrawal threshold; RT-qPCR, Quantitative polymerase chain reaction reverse transcription; SCDH, Spinal cord dorsal horn; SD, Sprague Dawley rats; ST36, Acupoints *Zusanli*.

## 1. Introduction

Neuropathic pain (NP) arises from pathologies or disorders impacting the somatosensory system, affecting over 10% of the global population [1]. This condition significantly impairs the mental health and social functioning of patients, thereby diminishing their quality of life and imposing a substantial burden on individuals, families, and society. However, despite the vast number of patients affected, current clinical interventions are largely ineffective in alleviating the clinical manifestations of NP in the majority of cases. For example, commonly prescribed opioids and non-steroidal anti-inflammatory analgesics exhibit minimal therapeutic efficacy for NP and frequently induce various adverse reactions [2]. Following nervous system injury, glial cells, particularly microglia and astrocytes, suffer from abnormal stimulation, predominantly in the spinal cord dorsal horn (SCDH) and dorsal root ganglion (DRG). These alterations may enhance the excitability of nociceptive sensory neurons, inducing peripheral and central sensitization. These neurons play a pivotal role in neural plasticity and contribute to the initiation and persistence of NP [3,4]. The pathophysiological mechanism underlying NP are intricate and no definitive treatment currently exists [5]. Moreover, the pharmacological management of NP is associated with adverse reactions. Electroacupuncture (EA) emerges as a safe and straightforward treatment option for NP, devoid of adverse reactions, and has demonstrated clinical efficacy in ameliorating the disease; however, the precise mechanism of action requires further investigation [6,7].

Research has demonstrated that an upregulation of DNA methyltransferase 3 Alpha (DNMT3a) expression within the DRG of NP animal models results in elevated methylation levels at the promoter regions of Oprm1 and Kcna2 genes. This, in turn, inhibits the expression of μ-opioid receptor (MOR) and potassium voltage-gated channel subfamily A member 2 (Kv1.2), thereby contributing to the development of hyperalgesia. If the increase of DNMT3a is reversed, hyperalgesia can be alleviated [8,9]. Furthermore, peripheral nerve injury reduces MOR expression in the sciatic nerve, DRG, and SCDH, causing hyperalgesia in rats [10,11]. Conversely, overexpression MOR in peripheral afferent nerves reduces hyperalgesia and enhances opioid analgesia [12]. Additionally, according to Du et al [13]. EA can enhance MOR expression in the DRG of the NP animal model.

Our previous study has demonstrated that EA could increase MOR expression in the gastrointestinal tract tissues. However, the potential of EA to modulate MOR expression in alleviating neuropathic pain, as well as the precise role and underlying mechanism of the DNMT3a/MOR signaling pathway regulated by EA, remain areas ripe for further exploration. The present innovative investigation was specifically designed to meticulously analyze the alterations in DNMT3a and MOR expression within in the spinal cord tissue along with DRG of rats subjected to chronic constriction injury (CCI), both before and after EA stimulation. Furthermore, this study employed a approach by administering adeno-associated virus (AAV) locally into either the spinal cord or DRG to either overexpress or silence DNMT3a, thereby enabling an in-depth examination of DNMT3a's function. The impact of DNMT3a/

MOR on neuropathic pain in CCI model rats was comprehensively evaluated through the observation of changes in pain behavior, pain-related factors (Fos, ERK1/2, and pERK1/2), and glial cell activation (CD11b, Iba1, and GFAP), thereby shedding new light on the potential therapeutic implications of EA in the management of neuropathic pain.

## 2. Materials and methods

### 2.1. Animal preparation

In the experiment, 48 male Sprague Dawley (SD) rats weighing between 225–250 g, were sourced from the Southern Medical University Laboratory Animal Center in Guangzhou, China. The animal experiments were granted approval by the Animal Ethics Committee of the Guangdong Academy of Traditional Chinese Medicine (ethics number 2020048) in accordance with the National Research Council's Guide for the Care and Use of Laboratory Animals. The animals were kept in the Guangdong Academy of Chinese Medicine Laboratory Animal Center and subjected to a 12-h light-dark cycle in conjunction with adequate water and food that could be freely obtained. The experimental procedures complied with the welfare of laboratory animals regulations. During behavioral tests, the grouping and treatment were kept confidential.

The experiment was completed in two parts. In the first part, the male SD rats were randomly divided into sham CCI group (Sham, n = 8) and CCI group (n = 16). The models of hyperalgesia were established in CCI rats.The CCI rats were then randomly divided into CCI group (CCI, n = 8) and electroacupuncture-treated group (EA, n = 8). In the second part, the male SD rats were randomly divided into normal control group (NC, n = 8) and DRG AAV injection group (n = 16). AAV injection group rats were injected with AAV overexpressing DNMT3a into the DRG. DRG AAV injected rats were then randomly divided into DRG AAV injection group (AAV, n = 8) and DRG AAV injection/electroacupuncture-treated group (AAV + EA, n = 8).

### 2.2. Animal model

The CCI model was developed according to previous literature. In simple terms, the rats were anesthetized using isoflurane, and the muscle was bluntly separated, and then four loose knots on the sciatic nerve trunk were ligated with a 4−0 suture. Subsequently, the muscle and skin were sutured together. The Sham group was subjected to the same as the CCI group, except that it did not undergo ligation. Animals with impaired motor function and death were excluded.

### 2.3. Pain behavioral test

The Von Frey Filament was utilized to induce mechanical paw withdrawal threshold (PWT), and then measured by the up-down technique (Aesthesio®, Danmic Global, San Jose, CA, USA). Dixon's formula was employed to ascertain the mechanical PWT of rats, necessitating the conversion of both positive and negative reaction patterns to a threshold of 50%. The Ugo Basile thermal pain tester (Ugo Basile 37370, Italy) was utilized to determine the rat paw withdrawal latency (PWL). The infrared heat radiation intensity setting was fixed. To prevent any potential harm to the tissue, set a cut-off time of 20 s. Each plantar test of the paw was performed three times, each time more than 6 min apart.

### 2.4. DRG AAV injection

Rats were intraperitoneally injected with 50 mg/kg of 2% sodium pentobarbital for anesthesia. The hair around the spinal column of the back of rats was cut off, and the skin of the back was disinfected with iodophor. The skin was incised 1 cm from the upper edge of the median dorsal spine to passively separate the muscles and fascia on the right side of the spine to expose the lamina, facet and transverse process of the L4 and L5 Lumbar DRG. The L4 and L5 segments of the DRG were visible after the rats were fixed on a stereotaxic instrument, and the edge of the faceting process and lamina were opened using a microscope and a skull drill. Following the installation of the micro-injection pump and the glass electrode, the terminal end of the glass electrode was positioned at a 45° angle relative to the DRG. The injection depth

was maintained within a range of 0.2 to 0.3 mm, with an injection speed set at 30 nL/min. A total of $1.0 \times 10^{12}$ AAV particles were administered, and a dwell time of 5 min was observed post-injection. After the injection, the back muscles and skin of the rats were sutured layer by layer with a 4−0 suture, and the incision was sterilized with an iodophor. Finally, the rats were placed in a warm environment to wake up and placed in a cage for normal feeding.

## 2.5. Electroacupuncture (EA) treatment

EA treatment was performed 1 week after CCI and 3 weeks after AAV DRG injection. EA at acupoints *Zusanli* (ST36) and *Yanglingquan* (GB34) was performed 30 minutes per day for 7 days.Rats were maintained in isoflurane anesthesia with EA using an EA instrument (SDZ-III, 20/100 Hz, 1.5 mA). If a behavioral test and EA treatment were required on the same day, the behavioral test was performed 2 h after the end of EA treatment.

## 2.6. Western blotting

Rats were intraperitoneally injected with 2% pentobarbital sodium 150 mg/kg and euthanized by decapitation. Subsequently, the L3~5 DRGs and L3~5 spinal cords were collected. Tissues were ground in the lysate (containing protease and phosphatase inhibitors). The supernatant was separated at 4°C, 8,000 g for 30 min after centrifugation. The protein concentration was detected by BCA method. After adding a protein loading buffer, the sample was subjected to a temperature of 100°C for10 min. Proteins were transferred by electrophoresis onto a polyvinylidene fluoride (PVDF) membrane and sealed with 5% skim milk powder (dissolved in 0.1% TBST) for 1 h at room temperature. The membrane was incubated overnight at 4°C with primary antibodies containing: anti-MOR (rabbit, 1:2000, Bioss, bs-3623R), anti-DNMT3a (rabbit, 1:2000, Abcam, ab188470), anti-ERK1/2(rabbit, 1:1000, Cell Signaling Technology), anti-pERK1/2(rabbit, 1:1000, Cell Signaling Technology), anti-GAPDH (rabbit, 1:5000, Proteintech, 10494–1-AP), anti-α-Tubulin (rabbit, 1:2000, Proteintech, 11224–1-AP). After three washes with 0.1% TBST, the membrane was soaked in a secondary antibody (rabbit, 1:5000, Proteintech, SA00001–2), coupled with horseradish peroxidase for 1 h at room temperature. An enhanced developer was used to visualize the protein, and images were generated employing the ChemiDoc XRS system and Image Lab software. Image Lab software was used to analyze the grayscale band values and control GAPDH, and the values of α-Tubulin were homogenized.

## 2.7. Quantitative polymerase chain reaction reverse transcription (RT-qPCR)

RNA extraction from tissue samples was conducted via TRIzol reagent (Invitrogen; Thermo Fisher Scientific, Inc.), reverse transcription, and amplification via Evo M-MLV RT KIT II (Accurate Biotechnology Co, Ltd.), together with SYBR® Green Pro Taq HS Premix qPCR Kit I (Accurate Biotechnology Co, Ltd.). The protocol was performed according to the product instruction. The PCR reaction was repeated three times per trial. The Applied Biosystems, Inc.'s sequence detection system software was utilized. GAPDH acted as an internal standard control gene and target mRNA levels were evaluated from the quantitation cycle number. The primers were designed by NCBI Primer and synthesized by the company (Sangon Biotech (Shanghai) Co, Ltd. and Invitrogen). All primer sequences were shown in the attached Table 1.

## 2.8. Immunofluorescence staining

The rats were subjected to deep anesthesia using isoflurane and perfused 50 mL PBS by the ascending aorta, and then 100 mL of 4% paraformaldehyde was added. After perfusion, the DRGs located at L3-L5 and the spinal cord segments at L3-L5 levels were extracted and subsequently preserved in a 4% paraformaldehyde solution overnight at 4°C. Next day, the tissues were dehydrated by immersion in a solution containing 30% sucrose. The tissues were all immersed in the bottom of EP tube containing 30% sucrose solution and subsequently embedded with OCT. The thicknesses of the spinal cord segments and DRGs slices were 30 um and 10 um, respectively. The sections were subjected to a blocking

**Table 1. RT-qPCR primer sequences.**

| Gene | Primer | Primer Sequence (5'-3') |
|---|---|---|
| CD11b | Forward primer | 5'- GGAAGGTGTCAGCAAGCCAGAAC-3' |
| | Reverse primer | 5'- TTAGCGGGAAAGATGGGATGGTTTATG-3' |
| Iba1 | Forward primer | 5'-GATGATCCCAAGTACAGCAGTGATGAG-3' |
| | Reverse primer | 5'- AACCCCAAGTTTCTCCAGCATTCG-3' |
| GFAP | Forward primer | 5'- CAGACCTCACAGACGTTGCTTCC-3' |
| | Reverse primer | 5'- AGTTGGCGGCGATAGTCATTAGC-3' |
| c-fos | Forward primer | 5'- AGACCATGTCAGGCGGCAGAG-3' |
| | Reverse primer | 5'- GTCAGCTCCCTCCTCCGATTCC-3' |
| GAPDH | Forward primer | 5'-GAAGGTGAAGGTCGGAGTC-3' |
| | Reverse primer | 5'-GAAGATGGTGATGGGATTTC-3' |

procedure in a solution with 0.3% triton X-100 and 5% normal donkey serum for 1 h at room temperature, following the manufacturer's instructions. The primary antibodies were applied to the sections in a humidified chamber and left overnight at 4°C. On a subsequent day, the sections underwent multiple washes followed by 1 h incubation with the secondary antibody at room temperature. Then, the sections underwent multiple washes and were covered with a few drops of DAPI. The stained sections were examined and the images were captured with Nikon fluorescence microscopy. Images directly comparing different groups were taken using the same acquisition settings. All antibodies used in immunofluorescence experiments are as follows: anti-Iba1 (Rabbit, 1:1000, Wako, 019−19741), anti-GFAP (1:500, Sigma, SAB5700611), anti-MOR (Rabbit, 1:200, Bioss, bs-3623R), anti-DNMT3a (mouse, 1:200, Abmart, M30138S), Goat Anti-Rabbit IgG (H+L), CoraLite488 conjugate (1:200, Proteintech, SA000013−2), Goat Anti-Rabbit IgG (H+L), CoraLite594 conjugate (1:200, Proteintech, SA000013−4), Goat Anti-Mouse IgG (H+L), CoraLite594 conjugate (1:200, Proteintech, SA000013−3).

## 2.9. Statistical analysis

Statistical analysis was performed using IBM SPSS 20.0 software, while graphical plotting was executed using GraphPad Prism 8.0 software. The behavioral data underwent normal distribution analysis and a two-way ANOVA for analysis. Additional data were subjected to analysis through the employment of one-way ANOVA. Tukey's method was used for multiple comparisons between the groups. All data were presented in the form of mean values along with their corresponding standard errors. A level of statistical significance was supposed to a $P$-value of less than 0.05.

## 3. Results

### 3.1. EA alleviated hyperalgesia and hypersensitivity in CCI rats

Fig 1A depicted the flow chart of the experimental design. Before CCI surgery, there was no variation in baseline PWT and PWL among the three groups. Compared to the Sham group, rats that underwent CCI surgery exhibited a significant reduction in both PWT and PWL. After the administration of EA treatment, a notable increase in PWT (Fig 1B) and PWL (Fig 1D) was observed in the ipsilateral hindlimb of the EA cohort. The application of EA did not produce a significant impact on PWT (Fig 1C) and PWL (Fig 1E) of the contralateral hindlimb among rats with CCI.

### 3.2. EA reduced pain-related signals expression in CCI rats DRG and SCDH

Several pain-related signaling modifications accompany with nerve injury-induced NP. These signal alterations are the basis and manifestation of the development of peripheral and central sensitization, including transcription factors, signaling pathways, and activation of glial cells. Fos is an immediate early response gene that produces an early response to

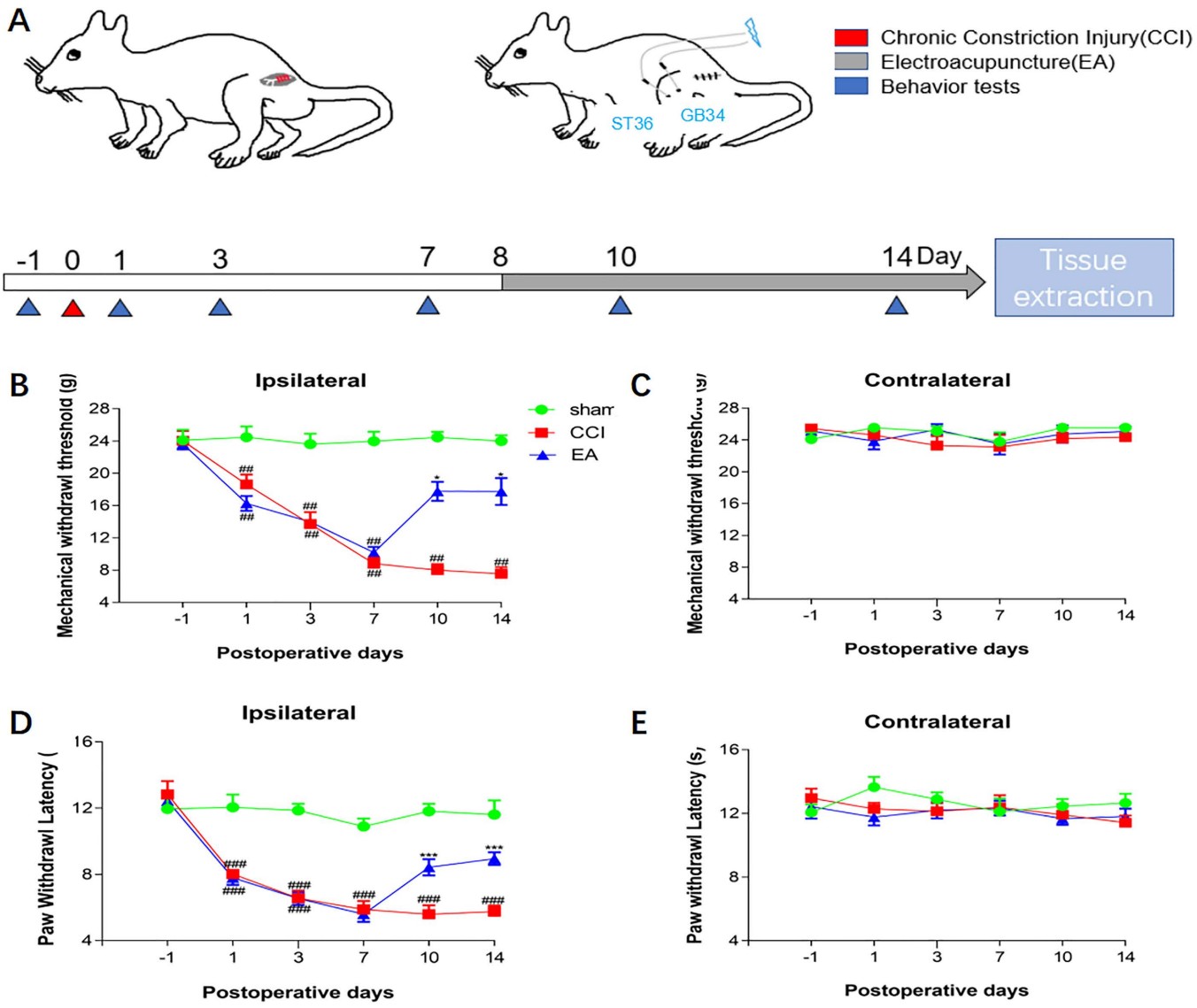

**Fig 1. EA alleviated hyperalgesia and hypersensitivity in CCI rats.** (A) Schematic diagram of rat intervention. (B–C) Changes of **MWT**(g) in normal rats, CCI rats, and rats with EA at different time points (1, 3, 7, 10, and 14 days) (n = 8). (D–E) Changes of **PWL**(s) in normal rats, CCI rats, and rats with EA at different time points (1, 3, 7, 10, and 14 days) (n = 8). Analyzed by normal distribution analysis and a two-way ANOVA for analysis. $^{\#}P$ <0.05, $^{\#\#}P$ <0.01, $^{\#\#\#}P$ <0.001.$^{*}P$ <0.05, $^{**}P$ <0.01, $^{***}P$ <0.001.MWT: Mechanical withdrawal threshold. PWL: Paw withdrawal latency. CCI: chronic constriction injury. EA: Electroacupuncture.

various cellular stimuli. The c-Fos encoded by the Fos gene is often used as a marker of neuronal activity. In this study, a significant elevation in Fos transcription levels was detected in the ipsilateral DRG of the CCI group, in contrast to the sham group (**Fig 2A**), and there was no significant increase in the EA group. The study results indicated that there was no statistically significant variation in Fos mRNA levels observed in the contralateral DRG between the three groups (**Fig 2B**). The transcription level of Fos in SCDH in the CCI group was as significantly elevated, similar to the ipsilateral DRG (**Fig 2C**). The EA group did not show a statistically significant increase in Fos mRNA levels. The results of Western blot analysis indicated a significant increase in the expression of pERK1/2 protein in the ipsilateral RDGs (**Fig 2D**–**2E**) and SCDH in

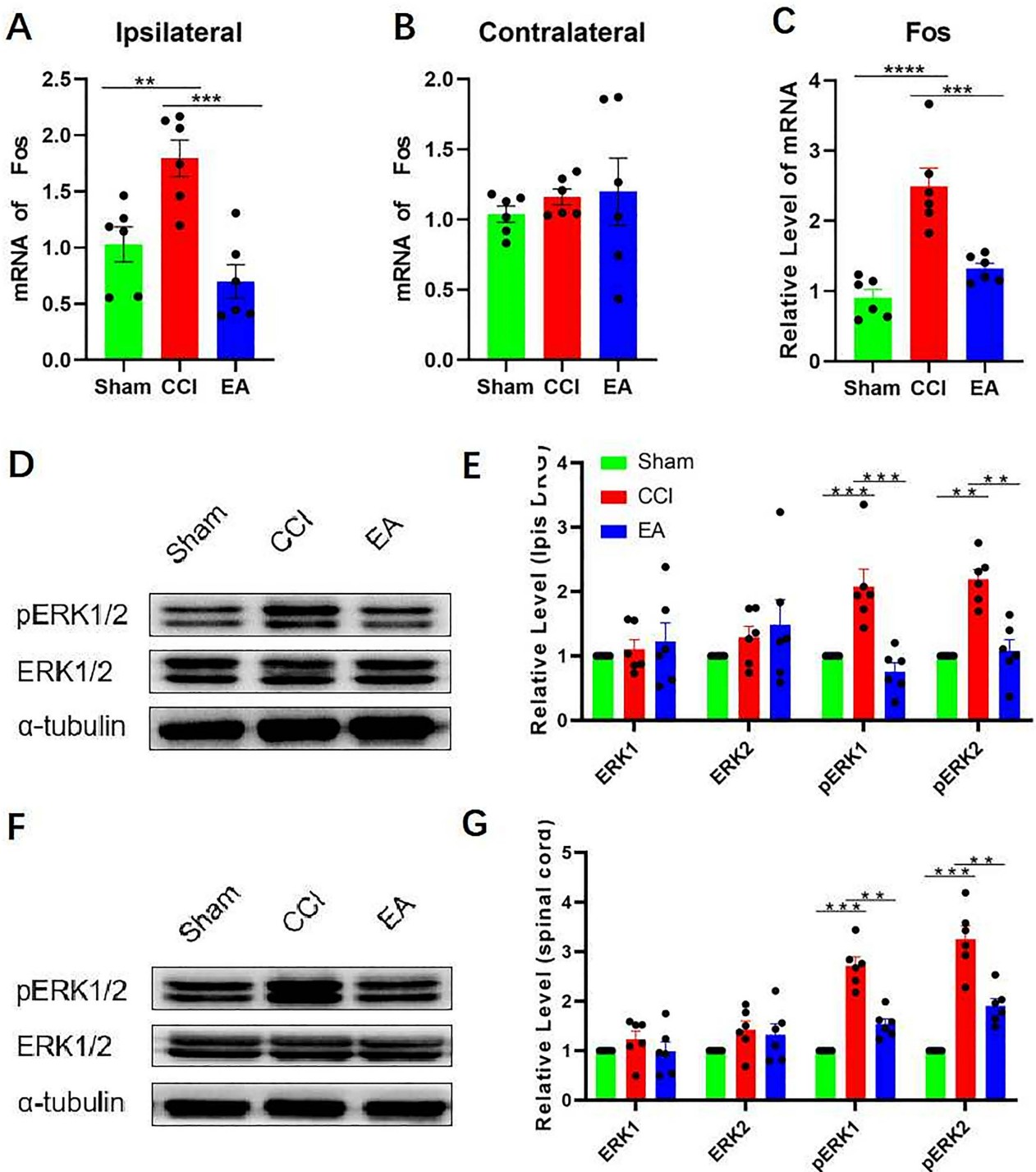

**Fig 2. EA reduced pain-related signals expression in CCI rats DRG and SCDH.** (A–C) RT-qPCR confirmed the mRNA expression of **Fos** in normal rats, CCI rats, and rats with EA. The data were normalized to GAPDH (n = 6). (D–E) Western blotting of the ERK1/2 and pERK1/2 protein expression in DRG of normal rats, CCI rats, and rats with EA. Data were normalized to α-tubulin and represented as mean ± SD (n = 6). (F–G) Western blotting of the ERK1/2 and pERK1/2 protein expression in SCDH of normal rats, CCI rats, and rats with EA. Data were normalized to α-tubulin and represented as mean ± SD (n = 6). Analyzed by one-way ANOVA and Tukey's method. *$P<0.05$, **$P<0.01$, ***$P<0.001$. RT-qPCR: Reverse transcription-Quantitative polymerase chain reaction. DRG: dorsal root ganglia. SCDH: spinal cord dorsal horn CCI: chronic constriction injury. EA: Electroacupuncture.

CCI group (**Fig 2F**–**2G**). pERK1/2 protein levels in DRG and SCDH were significantly reduced by EA treatment compared to the CCI group. The significance of neuropathic pain was demonstrated to be closely linked to the phosphorylation of ERK1/2.

Previous research on neuropathic pain extensively delved into the role of glial cells. Among them, microglia and astrocytes are strongly associated with neuropathic pain. To further determine the effect of EA on microglia and astrocytes, RT-PCR was used to detect common markers of both glial cells in ipsilateral DRG and SCDH. The present study revealed a significant elevation in mRNA levels of CD11b, Iba1, as well as GFAP in ipsilateral DRG in the CCI group (**Fig 3A**). However, the EA group did not change significantly. At the spinal cord level, we found similar changes in ipsilateral DRG (**Fig 3B**). Immunofluorescence results further confirmed that EA treatment could inhibit massive activation of microglia and astrocytes in SCDH (**Figs 3C**–**3E**).

### 3.3. EA inhibited the DNMT3a overexpression and restore MOR expression in the CCI rats DRG instead of SCDH

Compared to the CCI group, we found that EA treatment inhibited the increase of DNMT3a mRNA levels in ipsilateral DRG in rats after CCI surgery (**Fig 4A**). Compared to the sham group, the levels of MOR mRNA was decreased in the ipsilateral DRG of rats in the CCI group. The trend of change could be reversed after EA treatment (**Fig 4C**). In contralateral DRG, no significant variations were detected in DNMT3a and MOR mRNA levels among the three groups (**Figs 4B** and **4D**). The results of the Western blot analysis indicated a significant increase in DNMT3a expression levels in the ipsilateral DRG of rats after CCI surgery. However, no significant variations were detected between the EA and the sham groups (**Figs 4E**–**4F**). MOR expression levels decreased significantly after CCI surgery, while electroacupuncture could restore MOR expression (**Figs 4E** and **4G**). Immunofluorescence results showed increased DNMT3a expression and decreased MOR expression in ipsilateral DRG of rats in the CCI group (**Figs 4H**–**4J**). Electroacupuncture could reverse DNMT3a as well as MOR expression trends in ipsilateral DRG of rats in the CCI group.

To conduct a further investigation of the impact of EA on DNMT3a, as well as MOR at the spinal cord level, we examined the expression of DNMT3a and MOR in SCDH. RT-qPCR findings showed no difference in DNMT3a mRNA levels among the three groups (**Fig 5A**). In comparison to the sham group, the CCI group of rats did not exhibit a significant difference in MOR mRNA levels in SCDH. However, the EA group demonstrated significantly elevated MOR mRNA levels (**Fig 5B**). Compared to the sham group and the EA group, Western blot analysis indicated an elevation in the expression of DNMT3a in the CCI group, but it was not statistically significant (**Figs 5C**–**5D**). No significant variation in MOR expression was observed among the three groups (**Figs 5C** and **5E**). Finally, immunofluorescence showed no difference in DNMT3a and MOR expression in SCDH among the three groups (**Figs 5F**–**5H**).

### 3.4. EA relieved hyperalgesia and hypersensitivity caused by DNMT3a overexpression in DRG

To further determine the role of increased DNMT3a expression in hyperalgesia and hypersensitivity, DNMT3a was overexpressed by injecting AAV into DRG of rats. The experimental design flowchart was shown in **Fig 6A**. The AAV injection group of rats exhibited a statistically significant decrease in PWT and PWL compared to the NC group (**Fig 6B**–**6C**). Following electroacupuncture treatment, a significant increase in PWT, as well as PWL, was identified in the ipsilateral hind limbs of rats in the AAV + EA group (**Fig 6B**–**6C**). Western blot analysis revealed a statistically significant increase of DNMT3a expression and a significant decrease of MOR expression in rat DRG of the AAV group, and this trend could be reversed by electroacupuncture (**Fig 6D**–**6F**).

## 4. Discussion

We investigated the effects of EA stimulation at ST36 and GB34 acupoints on nociceptive sensation in a rat model of chronic sciatic nerve compressive injury (CCI) and conducted an in-depth analysis of its underlying mechanisms. We found that EA stimulation at these two acupoints effectively alleviated pain behaviors in CCI model rats, an effect closely

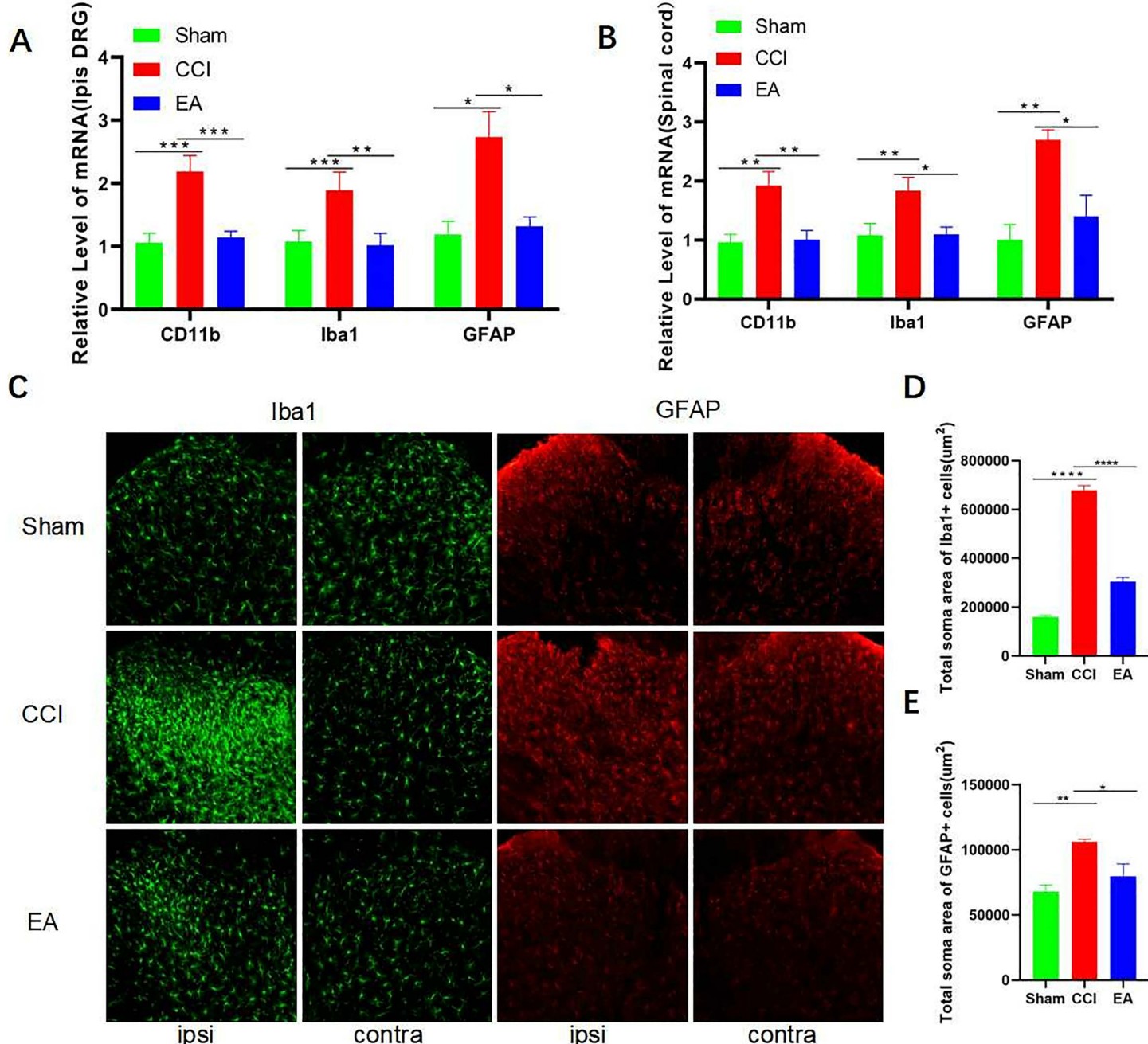

**Fig 3. EA inhibited massive activation of microglia and astrocytes in DRG and SCDH.** (A) RT-qPCR confirmed the mRNA expression of CD11b, Iba1, and GFAP **in DRG** of normal rats, CCI rats, and rats with EA. The data were normalized to GAPDH (n = 6). (B) RT-qPCR confirmed the mRNA expression of CD11b, Iba1, and GFAP in SCDH of normal rats, CCI rats, and rats with EA. The data were normalized to GAPDH (n = 6). Analyzed by one-way ANOVA and Tukey's method. (C–E) Iba1 and GFAP were observed under a fluorescence microscope in SCDH (n = 6). Scale bars: 100 μm. *$P < 0.05$, **$P < 0.01$, ***$P < 0.001$. RT-qPCR: Reverse transcription-Quantitative polymerase chain reaction. DRG: dorsal root ganglia. SCDH: spinal cord dorsal horn CCI: chronic constriction injury. EA: Electroacupuncture.

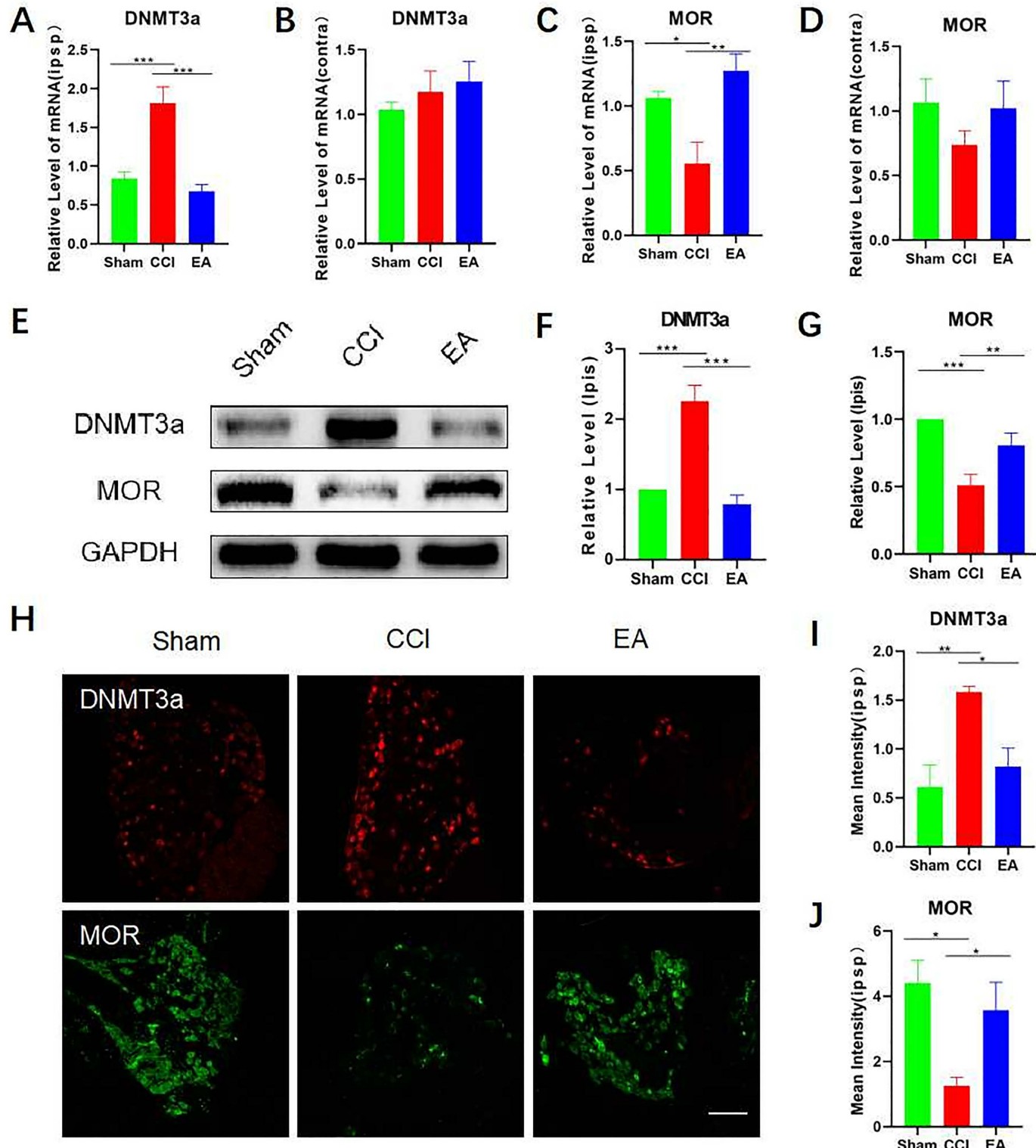

**Fig 4. EA inhibited the DNMT3a overecpression and restore MOR expression in the CCI rats DRG.** (A–B) RT-qPCR confirmed the mRNA expression of DNMT3a in DRG of normal rats, CCI rats, and rats with EA. Data were normalized to GAPDH (n = 6). (C–D) RT-qPCR confirmed the mRNA expression of MOR in DRG of normal rats, CCI rats, and rats with EA. Data were normalized to GAPDH (n = 6). (E–G) Western blotting of

DNMT3a and MOR in DRG of normal rats, CCI rats, and rats with EA. Data were normalized to GAPDH (n = 6). (H–J) DNMT3a and MOR were observed under a fluorescence microscope in DRG (n = 6). Scale bars: 100 μm. Analyzed by one-way ANOVA and Tukey's method. *$P < 0.05$, **$P < 0.01$, ***$P < 0.001$. RT-qPCR: Reverse transcription-quantitative polymerase chain reaction. DRG: dorsal root ganglia. CCI: chronic constriction injury. EA: Electroacupuncture.

associated with the modulation of DNMT3a and MOR signaling pathways in dorsal root ganglia (DRG). This finding aligns with previous research results, confirming the efficacy of EA in alleviating various pain conditions, including neuropathic pain. By measuring PWT and PWL in rats, we observed that EA stimulation significantly increased PWT and PWL in CCI rats, indicating a notable therapeutic effect of EA on CCI-induced hyperalgesia. Notably, the analgesic efficacy of EA stimulation at ST36 and GB34 was verified across multiple time points, suggesting a sustained and stable analgesic effect of EA. This discovery provides compelling evidence for the application of EA in chronic pain management.

The DRG serves as a crucial relay station for the transmission of nociceptive signals from the periphery to the central nervous system. In the CCI model, DRG neurons undergo a series of morphological and functional changes, including increased neuronal excitability, alterations in ion channel expression, and the release of neurotransmitters and neuromodulators. These changes collectively contribute to the abnormal transmission of nociceptive signals and the generation of pain sensation. The present study found that EA stimulation at ST36 and GB34 can regulate the expression of DNMT3a and MOR in DRG, thereby alleviating pain sensation in CCI model rats. This finding further confirms the pivotal role of DRG in pain modulation and unveils a novel mechanism by which EA exerts its analgesic effect through modulating DRG function.

In recent years, numerous studies have demonstrated the significance of epigenetic mechanisms in the modulation of pain-associated inflammatory conditions at the dorsal root and spinal levels. These remain the most extensively described functions of epigenetic mechanisms in the pathogenesis of pain conditions [14]. The involvement of epigenetic modifications, including DNA methylation, histone modifications, as well as non-coding RNA changes, in the pathogenesis and maintenance of neuropathic pain has been progressively recognized. DNA methylation is an important form of epigenetic modification, which is involved in a variety of biological processes by influencing gene expression. DNMT3a is one of the key enzymes in the process of DNA methylation, which plays an important role in regulating gene expression. The process of DNA methylation is facilitated by DNA methyltransferases (DNMTs), which can be classified into two distinct categories: DNMT1, responsible for maintaining DNA methylation, and DNMT3a/b, involved in generating DNA methylation [15]. Downregulation of DRG voltage-gated potassium channels induced by nerve injury is essential in neuronal excitability and hyperalgesia. Several studies confirm that DNMT1 and DNMT3a are involved in downregulating potassium channels in injured nerves in NP models [8,16,17]. Additionally, MOR expression is regulated by DNA methylation. Increasing of Oprm1 and Kcna2 gene expression in DRG neurons can be achieved by recruitment of DNA methyltransferase DNMT3a into gene promoters through the action of MDB1 (methyl-CpG binding domain protein 1). By inhibiting the expression of DNMT3a or MDB1, and Kv1.2, MORs can be restored, and neuropathic pain can be alleviated [16,18]. In the present study, we observed a significant upregulation of DNMT3a expression in the dorsal root ganglion (DRG) of rats subjected to the chronic sciatic nerve compressive injury (CCI) model. This upregulation may be associated with the neuropathic pain induced by CCI. Following electroacupuncture (EA) stimulation at ST36 and GB34, the expression of DNMT3a in the DRG was significantly downregulated, suggesting that EA may alleviate pain by inhibiting the activity of DNMT3a. However, the precise mechanism of DNMT3a in pain modulation remains incompletely understood.

Acupuncture is a promising adjunctive therapy for the NP. Compared to opioids, acupuncture analgesia is more effective, faster, better tolerated, and has fewer side effects. Studies show that acupuncture as a non-drug therapy can effectively reduce opioid use and dependence [19,20]. The EA therapeutic apparatus combines acupuncture techniques with the application of stable frequency pulse current for electrical stimulation. This involves the insertion the filiform needle into specific potints on the body. Because EA does not require regular manual needle stimulation, it may be more effective

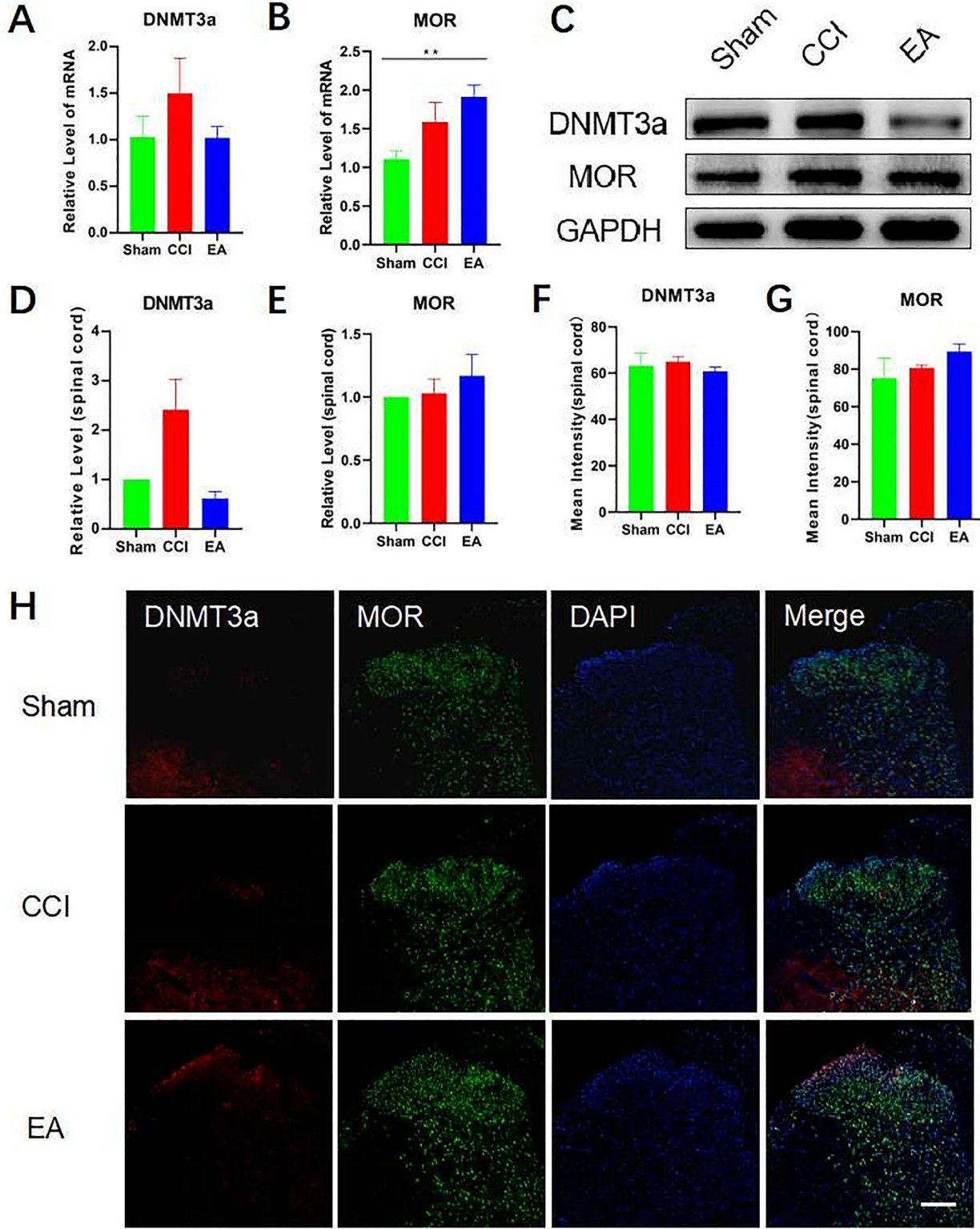

**Fig 5. EA did not influence DNMT3a and MOR expression in the CCI rats SCDH.** (A) RT-qPCR confirmed the mRNA expression of **DNMT3a** in SCDH of normal rats, CCI rats, and rats with EA. Data were normalized to GAPDH (n = 6). (B) RT-qPCR confirmed the mRNA expression of **MOR** in SCDH of normal rats, CCI rats, and rats with EA. Data were normalized to GAPDH (n = 6). (C–E) Western blotting of **DNMT3a** and **MOR** in SCDH of

normal rats, CCI rats, and rats with EA. Data were normalized to GAPDH (n = 6). (H–J) **DNMT3a** and **MOR** were observed under a fluorescence microscope in SCDH (n = 6). Scale bars: 100 μm. Analyzed by one-way ANOVA and Tukey's method. *$P < 0.05$, **$P < 0.01$, ***$P < 0.001$. RT-qPCR: Reverse transcription-Quantitative polymerase chain reaction. DRG: dorsal root ganglia. SCDH: spinal cord dorsal horn. CCI: chronic constriction injury. EA: Electroacupuncture.

in treating chronic pain than manual needle manipulation [21]. During the treatment, the stimulation's frequency, intensity, and duration remained constant. In this work, EA was employed to reduce pain sensitivity in CCI rats at *Zusanli* (ST36) and *Yanglingquan* (GB34). According to Wang et al. [22], ST36 is the "Lower Hesea" point and the "He-sea" point of the Stomach Meridian of the Foot-Yangming [22]. GB34 is a part of the Foot-Shaoyang gallbladder meridian, which connects the tendon points of the eight meridians with the gallbladder meridian's lower joint point. GB34 has been found to play a role in anti-inflammation, analgesia, and synaptic remodeling in the peripheral and central nervous systems, respectively. The two points can work together, relieving pain and discomfort induced by nerve injury.

MOR serves as the primary target for opioids and is a vital component of the endogenous analgesic system. In pain modulation, MOR inhibits neuronal excitability by activating downstream signaling pathways, thereby alleviating pain sensation. An increasing number of studies have indicated that dysfunction of MOR is closely associated with various pain conditions. In the present study, we found a significant downregulation of MOR expression in the DRG of rats subjected to the CCI model. This downregulation may be related to the neuropathic pain induced by CCI. Following EA stimulation at ST36 and GB34, the expression of MOR in the DRG was significantly upregulated, suggesting that EA may alleviate pain by enhancing the function of MOR.

In the present study, mechanical and thermal nociceptive behavioral tests demonstrated that EA at ST36 and GB34 significantly improved pain sensitivity in rats with CCI of the sciatic nerve. RT-qPCR, western blot, and immunofluorescence techniques were employed to identify changes in Fos, ERK1/2, pERK1/2, Iba1, CD11b, GFAP, glial cell activation, DNMT3a, and MOR in the SCDH and DRG. Among them, increased Fos mRNA suggests possible neuronal activation, which needs further protein validationthe. In CCI rats, upregulation of DNMT3a expression in the DRG, along with upregulation of Oprm1 and Kcna2 gene expression, and downregulation of Kv1.2 and MOR expression, led to hyperalgesia and allodynia, promoting the occurrence of pain. Following EA treatment, downregulation of Fos, pERK1/2, Iba1, CD11b, and GFAP gene expression in both the DRG and SCDH resulted in increased paw withdrawal threshold (PWT) and paw withdrawal latency (PWL), exerting an analgesic effect (Fig 7).

Previous studies have shown that exogenous opioids (such as morphine) produce analgesic effects by acting on MOR in glutamatergic (vglut2+) neurons, and the endogenous opioid is released from the body alleviates chronic inflammatory pain by acting on MOR in GABA-inhibitory neurons (CFA and formalin injection models). [23] The activation of OCT1 has been observed to increase the expression of DNMT3a, but not DNMT3b, in DRG neurons following peripheral nerve injury. Blocking this change can prevent DNA methylation in the promoter region of the MOR gene and restore expression to its normal levels [8,9,24]. Our results showed that EA significantly inhibited DNMT3a expression in ipsilateral DRG of CCI rats and exhibited a partial reversal of the declined MOR expression in the DRG induced by CCI. Surprisingly, the alterations in DNMT3a and MOR expression in SCDH, unlike in DRG, were not statistically different. We further confirmed this result by overexpressing DNMT3a in DRG.

This study investigated the impact of EA stimulation at ST36 and GB34 on pain perception in rats with CCI of the sciatic nerve and found that EA may alleviate pain by modulating the DNMT3a/MOR signaling pathway in the DRG. However, despite achieving certain results, this study has some limitations that need to be addressed in future research. Firstly, there are limitations in sample selection. The study only used CCI model rats as subjects. Although the CCI model is commonly used in neuropathic pain research, pain mechanisms and responses may vary among different animal models. Therefore, future studies should consider using other animal models or multiple models for comparison to verify the universality and specificity of EA treatment for pain. Secondly, the limitations of the research methods cannot be ignored. This study mainly

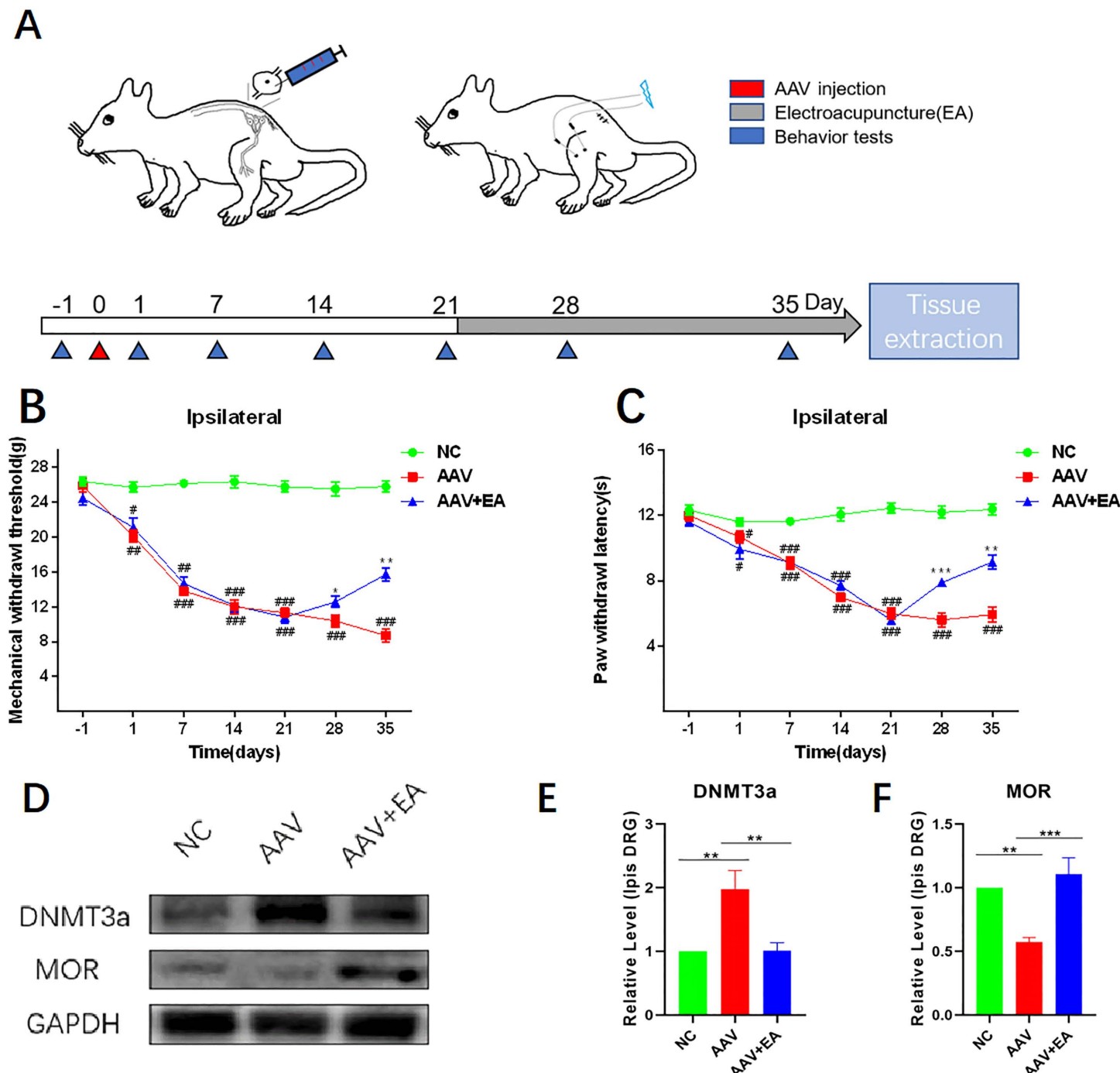

**Fig 6. EA relieved hyperalgesia and hypersensitivity caused by DNMT3a overexpression in DRG.** (A) Schematic diagram of rat intervention. (B) Changes of MWT (g) in normal rats, AVV rats, and rats with EA at different time points (1, 7, 14, 21, 28, 35 days) (n = 8). (C) Changes of PWL(s) in normal rats, AVV rats, and rats with EA at different time points (1, 7, 14, 21, 28, 35 days)(n = 8). Analyzed by normal distribution analysis and a two-way ANOVA for analysis. (D, E, F) Western blotting of DNMT3a and MOR in the DRG of normal rats, AAV rats, and rats with EA. Data were normalized to GAPDH (n = 8). Analyzed by one-way ANOVA and Tukey's method. #$P < 0.05$, ##$P < 0.01$, ###$P < 0.001$.*$P < 0.05$, **$P < 0.01$, ***$P < 0.001$. MWT: Mechanical withdrawal threshold. PWL: Paw withdrawal latency. EA: Electroacupuncture.

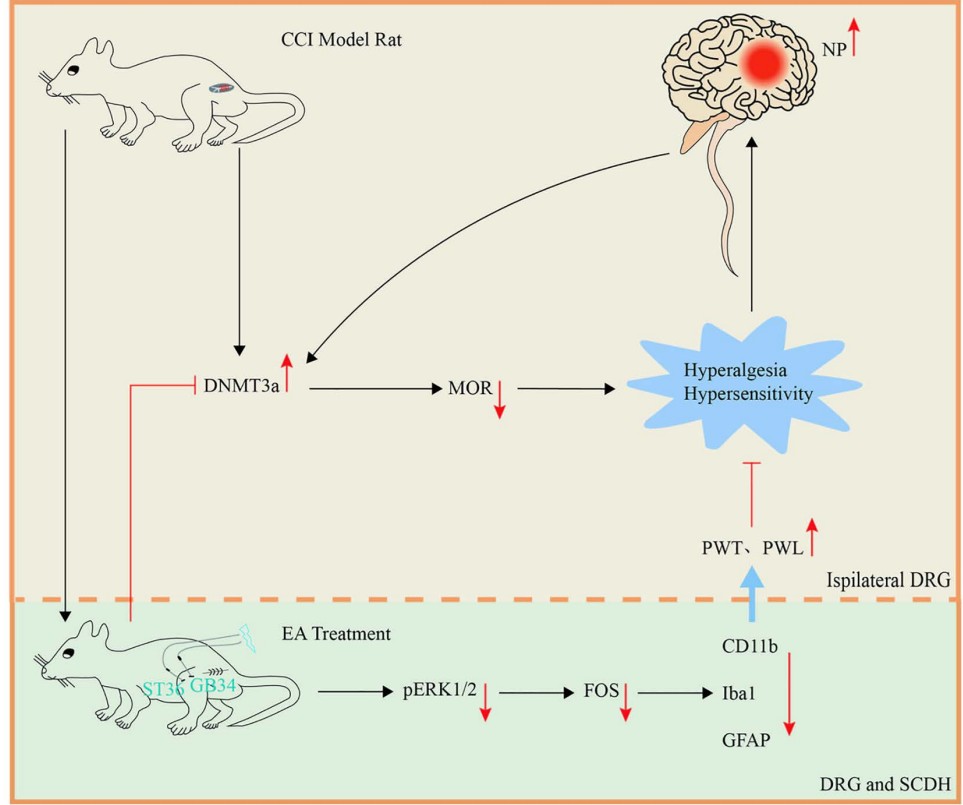

**Fig 7. The mechanism of EA therapy to exert analgesia.** DRG: dorsal root ganglion. CCI: chronic constriction injury. MOR: mu opioid receptor. SCDH: spinal cord dorsal horn. PWT: paw withdrawal threshold. PWL: paw withdrawal latency.

employed RT-qPCR, western blot, and immunofluorescence techniques to detect changes in the expression of relevant genes and proteins in the DRG and SCDH. Although these methods are widely used in molecular biology research, they primarily reflect static molecular-level changes and are difficult to comprehensively capture the dynamic changes during EA treatment. Therefore, future studies should consider adopting more real-time and dynamic detection methods, such as in vivo imaging, to more accurately assess the efficacy of EA treatment. Similarly, the current detection method cannot localize the specific activated cell types or spatial distribution. Additionally, this study has certain limitations in explaining the mechanism of EA treatment for pain. Although this study found that EA may alleviate pain by modulating the DNMT3a/MOR signaling pathway, the specific role and regulatory mechanisms of this pathway require further in-depth research. For example, the interaction between DNMT3a and MOR, how EA affects the expression of these molecules, and how these changes ultimately influence pain perception need to be elucidated in future studies. Finally, there are also limitations in the establishment of animal models and behavioral testing in this study. Although the establishment of CCI models and behavioral testing can simulate human neuropathic pain to some extent, there are still certain differences from the actual situation of human pain. Therefore, future studies should consider using animal models that are closer to human pain or combining clinical data for analysis to more accurately assess the clinical efficacy and safety of EA treatment for pain.

## 5. Conclusions

To sum up, we conclude that EA can reduce pain sensitivity by down-regulating the expression of DNMT3a in DRG, thereby improving MOR expression and distribution in DRG. The effect of EA on the DNMT3a/MOR signaling pathway

may be the possible mechanism of EA analgesia in CCI rats. This discovery provides compelling evidence for the application of EA in the management of neuropathic pain and reveals a novel mechanism by which EA exerts its analgesic effects by regulating DRG function.

## Supporting information

**S1 Data. Summary of raw data.** (https://figshare.com/articles/dataset/30071338).
(XLSX)

**S2 Data. Western blot raw data.** (https://figshare.com/articles/dataset/30071338).
(ZIP)

## Acknowledgments

We would like to acknowledge the reviewers for their helpful comments on this paper.

## Author contributions

**Conceptualization:** Feng Wang, Xiangyu Li, Gaofeng Zhao.

**Data curation:** Feng Wang, Chengcheng Zhou.

**Formal analysis:** Feng Wang, Chengcheng Zhou.

**Methodology:** Chuangbo Xie, Xiangyu Li.

**Software:** Haoyuan Wang.

**Validation:** Haoyuan Wang, Gaofeng Zhao.

**Writing – original draft:** Feng Wang.

**Writing – review & editing:** Chuangbo Xie, Gaofeng Zhao.

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
