## [Decision Letter · Decision Letter 0]

25 Jul 2025

PONE-D-25-32780Electroacupuncture Alleviates Pain-Like Behaviors Through Modulating DNMT3a/MOR Signaling Pathway in CCI RatsPLOS ONE

Dear Dr. Zhao,

Thank you for submitting your manuscript to PLOS ONE. After careful consideration, we feel that it has merit but does not fully meet PLOS ONE’s publication criteria as it currently stands. Therefore, we invite you to submit a revised version of the manuscript that addresses the points raised during the review process.

**Thank you for submitting the following manuscript to PLOS ONE.**

**Please revise the manuscript according to the reviewers' comments and upload the revised file.**

We look forward to receiving your revised manuscript.

Kind regards,

Yung-Hsiang Chen, Ph.D.

Academic Editor

PLOS ONE

Journal Requirements:

4. In the online submission form, you indicated that data will be made available on request.

This work was supported by the Natural Science Foundation of Guangdong Province [grant number 2020A1515010956].

Additional Editor Comments :

Thank you for submitting the following manuscript to PLOS ONE.

Please revise the manuscript according to the reviewers' comments and upload the revised file.

Reviewers' comments:

Reviewer's Responses to Questions

**Comments to the Author**

1. Is the manuscript technically sound, and do the data support the conclusions?

Reviewer #1: Yes

Reviewer #2: Yes

2. Has the statistical analysis been performed appropriately and rigorously? 

Reviewer #1: Yes

Reviewer #2: Yes

3. Have the authors made all data underlying the findings in their manuscript fully available?

Reviewer #1: Yes

Reviewer #2: Yes

4. Is the manuscript presented in an intelligible fashion and written in standard English?

Reviewer #1: Yes

Reviewer #2: Yes

5. Review Comments to the Author

Reviewer #1: This research investigates the mechanism by which electroacupuncture (EA) at Zusanli (ST36) and Yanglingquan (GB34) alleviates neuropathic pain in CCI rats, finding that EA relieves pain-like behaviors by inhibiting DNMT3a expression and restoring MOR expression in the dorsal root ganglion (DRG). The study is well-designed and the results are clearly stated. I suggest to accept the manuscript for publication after clarifying the following minor questions.

Q1. In the Abstract, the description of the duration of EA treatment is somewhat ambiguous, and the wording should be revised to enhance readability. Also, the full name should be indicated when an acronym appears for the first time.

Q2. Add explanations of the other detected indicators (Fos, ERK1/2, pERK1/2, CD11b, Iba1, and GFAP) in the Introduction section. This will help readers understand the relevance of these indicators to neuropathic pain and the study's mechanistic framework, thereby enhancing the logical coherence of the introduction.

Reviewer #2: Dear Editor,

Thank you for the opportunity to review this manuscript entitled “Electroacupuncture Alleviates Pain-Like Behaviors Through Modulating DNMT3a/MOR Signaling Pathway in CCI Rats.” The authors investigated the potential mechanism by which electroacupuncture alleviates CCI-induced pain in rats, focusing on the DNMT3a/MOR signaling pathway. This study holds certain scientific value and research significance. Below are my detailed comments and suggestions for the authors and editors to consider:

1. Overall evaluation

This manuscript focuses on pain mechanisms induced by chronic constriction injury (CCI) in rats, exploring the therapeutic effect of electroacupuncture. The authors combine RT-qPCR, immunofluorescence staining, Western blot, and behavioral pain tests to investigate whether electroacupuncture may attenuate pain by downregulating DNMT3a expression and upregulating MOR expression in DRG. Overall, the manuscript has a relatively complete structure, and the experimental design is clear, with some data being convincing. However, several issues remain. I recommend that the authors revise by adding experimental details, improving image quality, and strengthening logical interpretation before the manuscript is considered for publication.

2. Major issues and suggestions

(1) In Figure 2, the authors examined the dynamic changes of Fos mRNA in DRG and SCDH under the CCI model, providing molecular-level evidence for activation of the pain pathway. Notably, they found that transcriptional levels were significantly increased on the ipsilateral side in the CCI group and that electroacupuncture intervention could inhibit this effect, which is a valuable finding. However, relying solely on Fos mRNA levels as evidence of neuronal activation has clear limitations:

① mRNA elevation does not necessarily translate into increased functional c-Fos protein expression, which is the widely accepted marker for neuronal activation.

② The current detection method cannot localize the specific activated cell types (neurons vs. glial cells) or spatial distribution (e.g., specific laminae of the spinal dorsal horn).

Therefore, I strongly recommend that the authors add a double immunofluorescence co-labeling experiment (e.g., c-Fos/NeuN) to confirm the inhibitory effect of electroacupuncture on pain-related neuronal activation, and to specify its location and quantity. If experimental conditions do not allow this, please address in the discussion section:

① clearly state that the conclusions are based only on transcriptional data;

② interpret the results cautiously (e.g., “Increased Fos mRNA suggests possible neuronal activation, which needs further protein validation”);

③ explicitly acknowledge the limitation of not detecting protein expression and cell-type localization.

(2) The fluorescence images in Figure 3C and Figure 4H are not sufficiently clear. Please provide higher-resolution or clearer images to ensure accurate representation of the results.

(3) In the mechanistic diagram (Figure 7), the authors propose that CCI can induce upregulation of DNMT3a expression in DRG, along with increased transcription of Oprm1 and Kcna2. However, no experimental data or figures supporting these two targets are provided in the manuscript. Please supplement with relevant data or figures. Additionally, the manuscript mentions a “cascade reaction,” but this should be substantiated experimentally; otherwise, the term should be revised or used more cautiously.

(4) The discussion section is somewhat lengthy and parts of it lack clear structure. It is recommended to restructure this section by adding subheadings or adjusting the logical flow to improve clarity and coherence.

3. Minor issues and suggestions

(1) Please add the full name of DNMT3a in the abstract to help readers better understand the study background.

(2) Figure 1 lacks a figure number in the manuscript. Please check and correct this. Also, the rat model illustration in Figure 1 appears rather rough; it is recommended to redesign it for better clarity and professionalism. In Figure 2, please add scatter points on the bar charts to indicate the sample size and distribution, which would enhance the transparency of the data.

(3) Figure 5E and Figure 5F lack serial number labels. Please check and revise accordingly.

4. Recommendation

Major revision (Revision required before reconsideration)

I hope these comments will help the authors to improve the manuscript. With additional experimental support and enhancements in figure presentation and logical structure, the study may reach the publication standard.

6. PLOS authors have the option to publish the peer review history of their article (what does this mean? ). If published, this will include your full peer review and any attached files.

**Do you want your identity to be public for this peer review?** For information about this choice, including consent withdrawal, please see our Privacy Policy .

Reviewer #1: No

Reviewer #2: No

---

## [Author Response · Author response to Decision Letter 1]

11 Sep 2025

Response to Reviewers

Dear Editor and Reviewers:

Thank you for giving us the opportunity to submit a revised draft of the manuscript " Electroacupuncture Alleviates Pain-Like Behaviors Through Modulating DNMT3a/MOR Signaling Pathway in CCI Rats" for publication in the PLOS ONE. We appreciate the time and effort that you and the reviewers dedicated to providing feedback on our manuscript and are grateful for the insightful comments on and valuable improvements to our paper. We have incorporated most of the suggestions made by the reviewers. Those changes are highlighted within the manuscript. Please see below, in blue, for a point-by-point response to the reviewers' comments and concerns.

Reviewers’ comments to the Authors:

Reviewer #1

Reviewers’ comment 1: In the Abstract, the description of the duration of EA treatment is somewhat ambiguous, and the wording should be revised to enhance readability. Also, the full name should be indicated when an acronym appears for the first time.

Author response: Added, thanks.

Modified text: EA treatment was performed at acupoints Zusanli (ST36) and Yanglingquan (GB34) after CCI 1 week and after AAV dorsal root ganglion (DRG) injection 3 weeks. EA was performed 30 minutes per day for 7 days.

Reviewers’ comment 2: Add explanations of the other detected indicators (Fos, ERK1/2, pERK1/2, CD11b, Iba1, and GFAP) in the Introduction section. This will help readers understand the relevance of these indicators to neuropathic pain and the study's mechanistic framework, thereby enhancing the logical coherence of the introduction.

Author response: Added, thanks.

Modified text: The impact of DNMT3a/MOR on neuropathic pain in CCI model rats was comprehensively evaluated through the observation of changes in pain behavior, pain-related factors (Fos, ERK1/2, and pERK1/2), and glial cell activation (CD11b, Iba1, and GFAP), thereby shedding new light on the potential therapeutic implications of EA in the management of neuropathic pain.

Reviewer #2

Reviewers’ comment 1: In Figure 2, the authors examined the dynamic changes of Fos mRNA in DRG and SCDH under the CCI model, providing molecular-level evidence for activation of the pain pathway. Notably, they found that transcriptional levels were significantly increased on the ipsilateral side in the CCI group and that electroacupuncture intervention could inhibit this effect, which is a valuable finding. However, relying solely on Fos mRNA levels as evidence of neuronal activation has clear limitations:

① mRNA elevation does not necessarily translate into increased functional c-Fos protein expression, which is the widely accepted marker for neuronal activation.

② The current detection method cannot localize the specific activated cell types (neurons vs. glial cells) or spatial distribution (e.g., specific laminae of the spinal dorsal horn).

Therefore, I strongly recommend that the authors add a double immunofluorescence co-labeling experiment (e.g., c-Fos/NeuN) to confirm the inhibitory effect of electroacupuncture on pain-related neuronal activation, and to specify its location and quantity. If experimental conditions do not allow this, please address in the discussion section:

① clearly state that the conclusions are based only on transcriptional data;

② interpret the results cautiously (e.g., “Increased Fos mRNA suggests possible neuronal activation, which needs further protein validation”);

③ explicitly acknowledge the limitation of not detecting protein expression and cell-type localization.

Author response: Thanks. We have accepted the reviewer's suggestions and clearly stated the limitations of this study in the discussion section of the manuscript, including the lack of detection of Fos protein and unclear localization of cell types.

Modified text: In the present study, mechanical and thermal nociceptive behavioral tests demonstrated that EA at ST36 and GB34 significantly improved pain sensitivity in rats with CCI of the sciatic nerve. RT-qPCR, western blot, and immunofluorescence techniques were employed to identify changes in Fos, ERK1/2, pERK1/2, lba1, CD11b, GFAP, glial cell activation, DNMT3a, and MOR in the SCDH and DRG. Among them, increased Fos mRNA suggests possible neuronal activation, which needs further protein validationthe.

Therefore, future studies should consider adopting more real-time and dynamic detection methods, such as in vivo imaging, to more accurately assess the efficacy of EA treatment. Similarly, the current detection method cannot localize the specific activated cell types (neurons vs. glial cells) or spatial distribution (e.g., specific laminae of the spinal dorsal horn).

Reviewers’ comment 2: The fluorescence images in Figure 3C and Figure 4H are not sufficiently clear. Please provide higher-resolution or clearer images to ensure accurate representation of the results.

Author response: Thanks. We have replaced the unclear images pointed out by the reviewer, and the clarity of the replaced images meets the publication requirements.

Reviewers’ comment 3: In the mechanistic diagram (Figure 7), the authors propose that CCI can induce upregulation of DNMT3a expression in DRG, along with increased transcription of Oprm1 and Kcna2. However, no experimental data or figures supporting these two targets are provided in the manuscript. Please supplement with relevant data or figures. Additionally, the manuscript mentions a “cascade reaction,” but this should be substantiated experimentally; otherwise, the term should be revised or used more cautiously.

Author response: Thanks.

Based on the reviewer's suggestions, we have adjusted the content of the mechanism diagram and removed the targets that were not experimentally validated in this study.

We have removed the inaccurate description of the term 'cascade reaction' mentioned in the article.

Modified text:

Reviewers’ comment 4: The discussion section is somewhat lengthy and parts of it lack clear structure. It is recommended to restructure this section by adding subheadings or adjusting the logical flow to improve clarity and coherence.

Author response: Thanks. According to your suggestion, I have made the modification in the discussion section.

Modified text:

Reviewers’ comment 5: Please add the full name of DNMT3a in the abstract to help readers better understand the study background.

Author response: Added, thanks.

Modified text: The expressions of DNA Methyltransferase 3 Alpha (DNMT3a), Mu opioid receptors (MOR), pain-related signals (Fos, ERK1/2, and pERK1/2) and glial cell activity-related factors (CD11b, Iba1, and GFAP) in spinal cord dorsal horn (SCDH) and DRG were detected by western blotting, quantitative polymerase chain reaction reverse transcription (RT-qPCR).

Reviewers’ comment 6: Figure 1 lacks a figure number in the manuscript. Please check and correct this. Also, the rat model illustration in Figure 1 appears rather rough; it is recommended to redesign it for better clarity and professionalism. In Figure 2, please add scatter points on the bar charts to indicate the sample size and distribution, which would enhance the transparency of the data.

Author response: Thanks.

Based on the reviewer's suggestion, we have checked and confirmed the numbering of Figure 1 in the manuscript.

The rat model in Figure 1 was personally designed and hand drawn by our doctoral team, reflecting our passion for this study and respect for PLOS ONE.

According to the reviewer's suggestion, we have added a scatter plot in Figure 2 to show the sample size and its distribution.

Reviewers’ comment 7: Figure 5E and Figure 5F lack serial number labels. Please check and revise accordingly.

Author response: Added, Thanks.

---

## [Decision Letter · Decision Letter 1]

2 Oct 2025

Electroacupuncture Alleviates Pain-Like Behaviors Through Modulating DNMT3a/MOR Signaling Pathway in CCI Rats

PONE-D-25-32780R1

Dear Dr. Zhao,

We’re pleased to inform you that your manuscript has been judged scientifically suitable for publication and will be formally accepted for publication once it meets all outstanding technical requirements.

Kind regards,

Yung-Hsiang Chen, Ph.D.

Academic Editor

PLOS ONE

Additional Editor Comments (optional):

Congratulations on the acceptance of your manuscript, and thank you for your interest in submitting your work to PLOS ONE.

Reviewers' comments:

Reviewer's Responses to Questions

**Comments to the Author**

1. If the authors have adequately addressed your comments raised in a previous round of review and you feel that this manuscript is now acceptable for publication, you may indicate that here to bypass the “Comments to the Author” section, enter your conflict of interest statement in the “Confidential to Editor” section, and submit your "Accept" recommendation.

Reviewer #2: All comments have been addressed

2. Is the manuscript technically sound, and do the data support the conclusions?

Reviewer #2: Yes

3. Has the statistical analysis been performed appropriately and rigorously? 

Reviewer #2: Yes

4. Have the authors made all data underlying the findings in their manuscript fully available?

Reviewer #2: Yes

5. Is the manuscript presented in an intelligible fashion and written in standard English?

Reviewer #2: Yes

6. Review Comments to the Author

Reviewer #2: All comments have been addressed, and the manuscript has been improved significantly. I have no more concerns.

7. PLOS authors have the option to publish the peer review history of their article (what does this mean? ). If published, this will include your full peer review and any attached files.

**Do you want your identity to be public for this peer review?** For information about this choice, including consent withdrawal, please see our Privacy Policy .

Reviewer #2: **Yes: ** Diansan Su

---

## [Editor Report · Acceptance letter]

PONE-D-25-32780R1

PLOS ONE

Dear Dr. Zhao,

I'm pleased to inform you that your manuscript has been deemed suitable for publication in PLOS ONE. Congratulations! Your manuscript is now being handed over to our production team.

Kind regards,

on behalf of

Dr. Yung-Hsiang Chen

Academic Editor

PLOS ONE